# Hepatocellular Carcinoma Recurrence and Mortality Rate Post Liver Transplantation: Meta-Analysis and Systematic Review of Real-World Evidence

**DOI:** 10.3390/cancers14205114

**Published:** 2022-10-19

**Authors:** Khalid I. Bzeizi, Maheeba Abdullah, Kota Vidyasagar, Saleh A. Alqahthani, Dieter Broering

**Affiliations:** 1King Faisal Specialist & Research Center, Riyadh 12713, Saudi Arabia; 2Salmaniya Medical Complex, Manama 323, Bahrain; 3Faculty of Medicine, Arabian Gulf University, Manama 329, Bahrain; 4University College of Pharmaceutical Sciences, Kakatiya University, Warangal 506009, India; 5Johns Hopkins University, Baltimore, MD 21218, USA

**Keywords:** HCC recurrence, liver transplant, AFP, Milan criteria

## Abstract

**Simple Summary:**

This is a systematic and meta-analysis study that looked at the hepatocellular carcinoma recurrence rate and its risk factors after liver transplantation. The recurrence rate, overall survival rate, and mortality rates in HCC patients post-liver transplantation remain relatively high. Significant regional differences exist in the prevalence of the recurrence, overall survival, and mortality rates. These findings will be of valuable guidance both for clinicians considering patients for an LT, and for providing tailored post-transplant HCC recurrence counselling to different populations with different risk levels.

**Abstract:**

Background: liver transplantation (LT) is the best curative option for eligible patients with hepatocellular carcinoma (HCC), however recurrence remains a major concern. This meta-analysis aimed to investigate the prevalence and risk factors of HCC recurrence. Methods: studies were selected using PubMed, Epistemonikas, and Google Scholar databases published from inception to 15 May 2022 and a meta-analysis of the proportions was conducted. Observational studies reporting the prevalence of recurrent HCC after an LT were included, with the analysis being stratified by an adherence to the Milan criteria (MC), geographical region, AFP levels, and donor type. Results: out of 4081 articles, 125 were included in the study. The prevalence of recurrent HCC was 17% (CI: 15–19). Patients beyond the MC were more likely to recur than patients within the MC. Asian populations had the greatest prevalence of HCC recurrence (21%; CI: 18–24), whereas North American populations had the lowest recurrence (10%; CI: 7–12). The mortality rate after HCC recurrence was 9%; CI: 8–11. North American populations had the greatest prevalence of mortality with 11% (CI: 5–17). Conclusions: the recurrence, overall survival, and mortality rates among patients with HCC post-LT remains high, with substantial differences between regions.

## 1. Introduction

Liver cancer currently ranks sixth in the list of the most common cancers worldwide and is the third leading cause of all cancer mortalities, making it a significant global health challenge. In 2020, the incidence of liver cancer exceeded 900,000 cases and there were more than 830,000 liver cancer-related deaths [1,2,3].

Liver transplantation (LT) is considered the best option as a curative treatment for eligible patients, with survival rates of approximately 70% in 5 years [4,5,6]. Hepatocellular carcinoma (HCC) recurrence post-transplantation remains a concern, with significant morbidity and mortality rates. Even with an adherence to the most stringent criteria for transplant listing, the recurrence of HCC post-transplantation is seen with rates ranging from 8 to 20% [7,8].

The release of cancer cells during surgery or the prevalence of occult metastases post-LT gives rise to HCC recurrence [9], which can be either intra-hepatic or distant. The most common sites for a recurrence are the lungs, lymph nodes, and bone, with the presence of vascular invasion and satellite nodules in the explant being a recognized risk factor, along with the quantity and size of the tumours [10]. The time of a recurrence onset varies and is broadly divided into early- or late-onset, with the latter occurring after 2 years from a transplantation. Early onset recurrence is associated with a poor prognosis and it is likely a result of pre-existing malignant cells at the time of transplantation [11].

Some of the oncological predictors of HCC recurrence are well known. Examples of such predictors include: evidence of vascular invasion, the tumour differentiation degree, and the size and number of nodules. Non-oncological risk factors that may be associated with HCC recurrence include: age, blood transfusion, a prolonged cold ischemia time (>10 h), and a warm ischemia time >50 min [10].

Given the significant morbidity and mortality associated with HCC recurrence post-LT, it is important to have a better understanding of post-LT HCC recurrence across a wide range of patient cohorts [12,13,14]. Although there are observational studies that have evaluated HCC recurrence, survival, and mortality rate after a liver transplantation, they lacked proper secondary evidence. Thus, with this background, we conducted a systematic review and meta-analysis to assess the overall prevalence of HCC recurrence, survival, and the mortality rate after a liver transplantation procedure. 

## 2. Methods

The current meta-analysis was performed according to the Preferred Reporting Items for Systematic Review and Meta-Analysis (PRISMA). The PRISMA checklist is a quality tool for including relevant information [15]. The review was not registered. 

### 2.1. Search Strategy

The literature search was conducted on PubMed, Epistemonikos, and Google scholar, from inception to 18 April 2022, with no date and language restrictions. The search utilized an extended combination of keywords and Medical Subject Headings (MeSH) including “Liver neoplasms”, “Biliary tract cancer”, “liver carcinoma”, “hepatoblastoma”, “Advanced stage”, “Metastatic”, “Stage-III, “Stage-IV”, “OS”, “PFS”, and “rate of recurrence”. We performed a separate search for the systematic reviews to compare the list of included studies with the studies retrieved from the searches. Complete details about the search terms used in various databases have been listed in Appendix A.

### 2.2. Study Eligibility Criteria

The eligibility criteria for the review were designed according to the PICOS (Population–Intervention–Comparator–Outcome–Study design) format. We included studies that met the following criteria: an observational study design such as a cross-sectional, case-cohort, and cohort study on patients with hepatocellular cancer who had undergone a liver transplant, irrespective of age; gender; ethnic, economic; or regional status, and reported any of the following outcomes: the rate of recurrence and mortality. Duplicate studies, abstracts, letters, editorials, conference proceedings, retracted publications, review articles, meta-analyses, non-population-based studies, and interventional studies were excluded.

### 2.3. Study Selection

Two authors (KB, VK) independently performed first-pass screening (FPS) by reviewing the titles and abstracts of all the records retrieved to identify articles that potentially met the predefined eligibility criteria. The full texts of the eligible titles were downloaded and reviewed independently by 2 authors (KB, VK) in the second-pass screening (SPS) to determine relevant inclusion in the final analysis. The discrepancies regarding the selection of studies between the 2 reviewers during the FPS and SPS were resolved through a discussion, consensus, or consultation with a third reviewer (MA).

### 2.4. Data Extraction and Management

Two authors (KB, VK) independently extracted data from the included studies using predesigned data extraction templates. Discrepancies during the data extraction were resolved through a discussion with a third reviewer (MA). The following details were extracted: the study identification, author details, year of publication, geographical region, study objectives, study design and settings, patient sampling, characteristics of the study population (the age range, mean age, sex, and comorbidities), measures, and main findings (the rate of recurrence and mortality).

### 2.5. Statistical Analysis

The estimates of the rate of HCC recurrence and mortality were presented as proportions (%) with 95% confidence intervals (CI). To calculate the pooled prevalence estimates of the outcome variables, we applied the regional population size weights. Heterogeneity and its significance among the studies were determined by using the I2 statistic (% residual variation due to heterogeneity). Tau2 (method of moments estimate of between-study variance) was used for each of the pooled estimates. I2-value ranges from 0 to 100% and is considered as no, low, moderate, and high when I2-values are 0%, 25%, 50%, and 75%, respectively [16]. Given that the heterogeneity was very high (95–99% inconsistency), a random effect DerSimonian–Laird model was used in all analyses [16].

With regard to substantial heterogeneity, we used stratified analyses based on the study-level characteristics, such as the geographical region, age, publication year, various socioeconomic status, and male percentage to investigate the source of the heterogeneity. Cochran’s Q test, the degree of freedom (df), and the *p*-value resulting from Cochran’s Q test were applied in the assessment of the interaction between each factor’s subgroups. A *p*-value of <0.10 was considered statistically significant for Cochran’s Q test [17]. 

Meta-regression analysis was performed based on the year of publication to know how the HCC prevalence trend was changed over the period. Publication bias that may occur as a result of small study effects was identified by applying the funnel plots in addition to both Egger’s and Begg’s tests.

The statistical analyses were performed by using version 16 MP of STATA software, (StataCorp, College Station, TX, USA).

## 3. Results

A primary search of the databases retrieved 4028 studies. After the duplicates were removed, 3937 unique studies were screened, including 53 studies from a bibliographic search in determining whether the inclusion criteria, as described in the methodology section, had been met. It was found that 3713 studies were not related to the study topic or met one of the exclusion criteria. Two hundred and twenty-four studies were further assessed for eligibility through a full text examination, and 99 were excluded for not meeting the inclusion criteria. Finally, one hundred and twenty-five studies met the pre-defined eligibility criteria. The reasons for the exclusion of each study are shown in Figure 1. The list of articles that are excluded at the FPS level (n = 99) due to various reasons is presented in Appendix A.

### 3.1. Characteristics of Included Studies

All the studies included were published between 1991 and 2022. Sample size varied on a regional basis from 4 to 18,406, making a total of 55,333 patients. Although most of the studies included both men and women, one study included solely men. The participants’ ages ranged from 11.5 to 63.4 years old. Most of the studies were conducted in Europe, followed by Asia, North America, South America, and Africa. One hundred and twenty-five studies reported the data on the prevalence of the recurrence rate, whereas 53 studies reported mortality due to the recurrence rate. The characteristics of the included studies are summarized in Appendix A.

### 3.2. HCC Recurrence

All the included studies reported a prevalence of the recurrence rate among 55,333 participants with HCC after a liver transplantation. The pooled analysis revealed that a prevalence of the HCC recurrence rate is 17% (n = 55,333, 95% CI: 15–19; I2 = 98.1%, *p* = < 0.001, τ2 = 0.01), shown in Table 1 and Figure 2. The subgroup analysis between the HCC recurrence as per the year of the study’s completion revealed a decreasing trend; however, it was not statistically significant.

### 3.3. Recurrence Rate Based on Geographical Region

Region-wise data showed a significant change in the prevalence of HCC after an LT among a population from across the regions of the globe, ranging from 10% (95% CI: 7–12, *p* = 0.000) in the South American region to 12% (95% CI: 4–20, *p* = 0.005) in the African region and 14% (95% CI: 12–16, *p* = 0.000) in the European region. Studies from North American (19%, 95% CI: 12–25, *p* = 0.000) and the Asian region reported a higher prevalence rate of HCC 21% (95% CI: 18–24, *p* = 0.000). Moreover, most of the studies were conducted in the European region. A significant heterogeneity across the regions was observed (Q = 26.4, df = 4; *p* = 0.00).

### 3.4. Recurrence Rate Based on Milan Criteria

Twenty-nine studies investigated the prevalence of HCC among patients with an LT, based on the Milan criteria. A total of 1673 individuals met the Milan criteria prior to an LT, with a pooled prevalence of an HCC recurrence of 44% (95% CI: 18–24), compared to a pooled prevalence of 47 (95% CI: 37–57 among patients (n = 511) who did not meet the Milan criteria (Figure 3). However, these differences between the patients within and beyond the Milan criteria were not statistically significant (Q = 3.4, df = 1; *p* = 0.07). 

### 3.5. Recurrence Rate Based on UCSF Criteria

A total of seven studies investigated the prevalence of HCC among patients with an LT, as per the UCSF criteria. A total of 74 individuals met the UCSF criteria prior to their transplantation, with a pooled prevalence of an HCC recurrence of 33% (95% CI: 18–48), compared to a pooled prevalence of 67% (95% CI: 51–82) among patients (n = 139) who did not meet the UCSF criteria. However, these differences between the patients within and beyond the UCSF criteria were statistically significant (Q = 35.4, df = 13; *p* = 0.00). 

### 3.6. Recurrence Rate Based on Type of Donor

A total of ten studies investigated the prevalence of HCC among patients with an LT as per the type of donor. The prevalence of HCC was compared between the living donor liver transplant (LDLT) and the deceased donor liver transplant (DDLT) and was found to be 19% (95% CI: 12–26) and 81% (95% CI: 77–85), respectively. However, there was a significant heterogeneity difference in the prevalence between the patients who underwent with the LDLT and DDLT type (Q = 167.6, df = 9; *p* = 0.00). 

### 3.7. Recurrence Rate Based on Countries Income

Countries of different socioeconomic levels wise data showed a substantial change in the prevalence of HCC after an LT, ranging from 12% (95% CI: 4–20, *p* = 0.005) in low-income countries to 16% (95% CI: 14–19, *p* = 0.000) in higher-income countries. Studies from the middle-income countries (19%, 95% CI: 15–24, *p* = 0.000) reported higher rates of a prevalence of HCC. Moreover, most of the studies were conducted in the high-income country’s region. When comparing the subgroups from various socioeconomic levels, no significant heterogeneity variation was observed in the prevalence of a recurrent HCC (Q = 2.63, df = 2; *p* = 0.27).

### 3.8. Recurrence Rate Based on Publication Year

Publication year wise data showed a substantial change in the prevalence of HCC after an LT, ranging from 15% (95% CI: 13–17, *p* = 0.000) in studies published between the period of 2010 and 2022 to 19% (95% CI: 14–24, *p* = 0.000) in studies published between the years of 2000 and 2009. Studies published between the years of 1990 and 1999 (23%, 95% CI: 13–32, *p* = 0.000) reported higher rates of the prevalence of HCC. Moreover, most of the studies were published between the years of 2010 and 2022. When comparing the subgroups of various periods, no significant heterogeneity variation was observed in the prevalence of a recurrent HCC (Q = 4.09, df =2; *p* = 0.13).

### 3.9. Recurrence Rate Based on Population Age

A total of one-hundred and seventeen studies reporting age wise data showed a notable change in the prevalence of HCC after an LT ranging from 16% (95% CI: 14–18, *p* = 0.000) in the population group with an age >50 years to 26% (95% CI: 17–34, *p* = 0.000) in the population group with an age <50 years. The majority of the studies published included patients >50 years. When comparing the subgroups of the population age groups with >50 years and <50 years, a significant heterogeneity variation was observed in the prevalence of a recurrent HCC (Q = 4.64, df = 1; *p* = 0.03).

### 3.10. Recurrence Rate Based on Pretransplant AFP

A total of 38 studies reported the data on the pre-transplant AFP levels, which were also used to stratify the prevalence of a recurrent HCC. A recurrent HCC occurred in 14% (95% CI: 11–17) of participants in studies where the mean AFP was less than 50 ng/mL, compared to studies with a mean AFP greater than or equal to 50 ng/mL, which found a pooled prevalence of 20% (95% CI: 14–26). When comparing the subgroups of AFP > 50 ng/mL and <50 ng/mL, there was a significant heterogeneity variation observed in the prevalence of a recurrent HCC (Q = 4.64, df = 1; *p* = 0.03).

### 3.11. Mortality Rate in HCC Patients after LT

A total of 53 studies, comprising 9959 participants, reported the mortality rate among HCC patients following a liver transplantation. The overall prevalence of the mortality rate was found to be 9% (95% CI: 8–11; I2 = 88.2%, *p* = 0.00, 2 = 0.00). North America had the greatest mortality rate (11% (5–17), followed by Asia (10% [95% 8–12]), and then Europe (10% [95% 7–12]). The relationship between an HCC recurrence and the study completion year demonstrated a decreasing trend in mortality, owing to HCC recurrence, however it was not statistically significant (*p* = 0.81). There was no difference in the prevalence rate of mortality due to HCC observed between the studies published in the middle- and high-income countries (9%), whereas no study was investigated in the low-income countries. The prevalence of the mortality rate doubled in the patients age group <50 years (18% [95% (6–30)] as compared to the patients age group >50 years (18% [95% (6–30)].

### 3.12. Publication Bias Assessment

Egger’s and Begg’s tests indicated a statistically significant publication bias for the HCC recurrence and its mortality rate estimates (Egger test: *p* = 0.0000 & 0.000, Begg’s test: *p* = 0.001 & 0.004, respectively). A visual examination of the funnel plots showed asymmetry and suggested a publication bias, as shown in Figure 4.

## 4. Discussion

We assessed the prevalence of the recurrence rate in participants with HCC after an LT through a comprehensive systematic review and meta-analysis, which reported a range between 15% and 19% in recurrence. The most recently published observational studies have also indeed demonstrated a similar trend in patients with HCC after an LT [18,19]. Moreover, the results of a recently published systematic review and meta-analysis also aligned with our findings [20,21,22].

Geographical region was identified as playing a major role in HCC recurrence rates, with a considerable variance ranging between 10% in South American studies and 21% in Asian studies. On the one hand, different ethnic groups and their genetic backgrounds might account for the higher HCC recurrence rates identified in the studies that were published in the Asian region; a cohort study carried out in the United States on around 400 patients with cirrhosis as a result of HCV infection found a four-fold increased risk of HCC in the Asian population, and a two-fold risk in African American men, when compared with Caucasians [23]. The risk factors, including diabetes mellitus, alcohol, hepatitis B infection, and non-alcoholic hepatic steatosis are well recognized as being major players for HCC and its recurrence in these regions [24].

AFP levels play a significant role in pre- and post-transplant outcomes. The current guidelines and recommendations indicate that pre-LT AFP levels provide a prognosis value for post-transplant outcomes, although a fixed threshold has not been established thus far given the broad discrepancies in the published literature to date [25]. Upon stratifying the analysis by mean AFP levels prior to the transplant, we identified a recurrence to be higher in those studies with a mean AFP of 50 ng/m. AFP levels have also been recommended as a predictor for a successful pre-transplant tumour size reduction, to which further investigation is essential [26].

Studies on HCC have highlighted a link between the level of income and recurrence and survival rates. Higher levels of income go hand-in-hand with improved healthcare access, which leads to an earlier diagnosis of cancer, underscoring the importance of comparing countries with various income levels [27]. Our analysis compared post-LT HCC recurrence in low-, middle-, and high-income countries using income level subgroups, as defined by the World Bank. We found that recurrence is higher among patients in middle-income than those in high-income countries. We hypothesize that better medical care in high-income countries may have resulted in disproportionately better rates of recurrence after an LT [27]. Our findings are also fairly aligned with a recently published systematic review conducted by Tan et al., who reported a 15% rate of HCC recurrence after an LT [22]. Further research into the link between personal income, access to healthcare, and HCC recurrence needs to be undertaken, acknowledging that there are income inequalities within countries, and that a country’s income is not representative of individual income [22].

Manzia TM et al. [28], in a systematic review, assessed the risk of immunosuppressives on malignancy post-liver transplantation. Malignant tumour recurrence and de novo malignancies are the most frequent cause of mortality in adult OLT recipients, with an incidence up to 26%. Calcineurin inhibitors (CNIs) seem to have a cancer-promoting influence while the mTORi might play a slight protective role, reducing the incidence of post-transplant malignancy. In an earlier study, Manzia et al. [29] demonstrated the feasibility of immunosuppression withdrawal in 40% and 60% of well-selected adult and paediatric liver transplant recipients. This is likely to be a beneficial factor in minimizing HCC recurrence post-transplant.

As for safety, the prevalence of mortality due to HCC recurrence after an LT was identified as 8–11%. This rate is particularly high in males, with a percentage greater than 50. This is not surprising given the fact that HCC significantly occurs more frequently in males. The reported male-to-female ratio of the incidence is at least 2:1, with reports as high as 4:1 across populations. 

Tan DJH et al. published a systematic review and meta-analysis in 2021 [22], including 58 studies focusing on HCC recurrence with 40,495 patients, to which we have expanded on and provided novel findings. Our study includes over one hundred and twenty-five full-text studies which has allowed us to achieve more consistent pooled estimates. Second, whereas a previously published systematic review only presented the results of HCC recurrence rate in LT patients, we consolidated the quantitative evidence on the wide-ranging impact of the rate of recurrence and mortality due to the recurrence in patients with HCC who underwent an LT in various geographical regions across the globe. Third, having identified the potential source of heterogeneity between the studies and subgroups, we further performed stratified meta-analyses. Nonetheless, the study findings are subject to the following limitations: first, the results may be influenced by widely varying factors such as cultures and practices in different geographic areas across the world, and it is important to note that more studies had been conducted in Europe than in any other region. Second, differences in sample size (varying between a few hundred to a few thousand) may account for the higher heterogeneity in the outcomes. A low power and precision may generate a higher Cochran q (heterogeneity x^2^ test statistics) and I2. Finally, we did not perform a quality assessment for the included studies due to the high number of studies which are included in this systematic review.

## 5. Conclusions

Almost half of HCC patients worldwide who undergo an LT are faced with a recurrence. Geographical region has been identified to play a major role in HCC recurrence, overall survival, and the mortality rates, making these findings of valuable guidance both for clinicians considering patients for an LT, and for providing tailored post-transplant HCC recurrence counselling to different populations with different risk levels. 

## Figures and Tables

**Figure 1 cancers-14-05114-f001:**
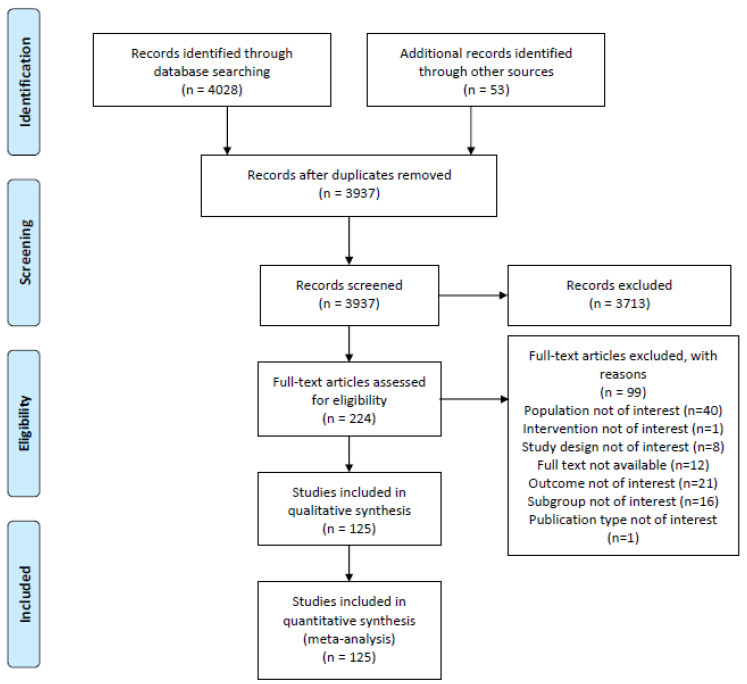
PRISMA diagram of the literature selection in this systematic literature review and meta-analysis.

**Figure 2 cancers-14-05114-f002:**
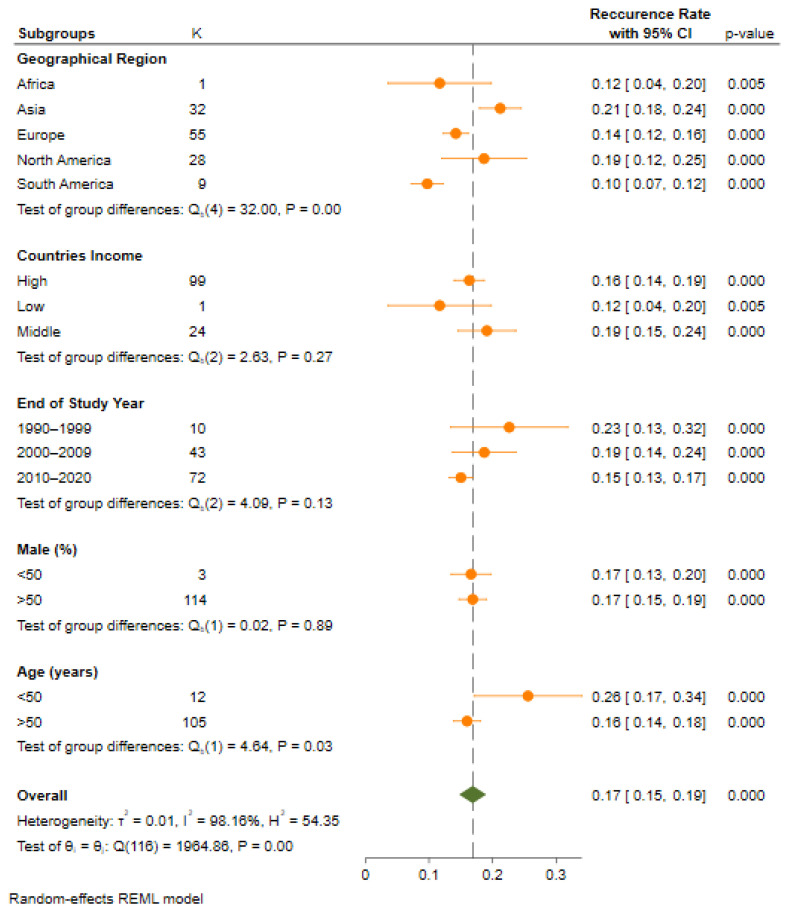
Pooled prevalence of HCC recurrence of all included studies.

**Figure 3 cancers-14-05114-f003:**
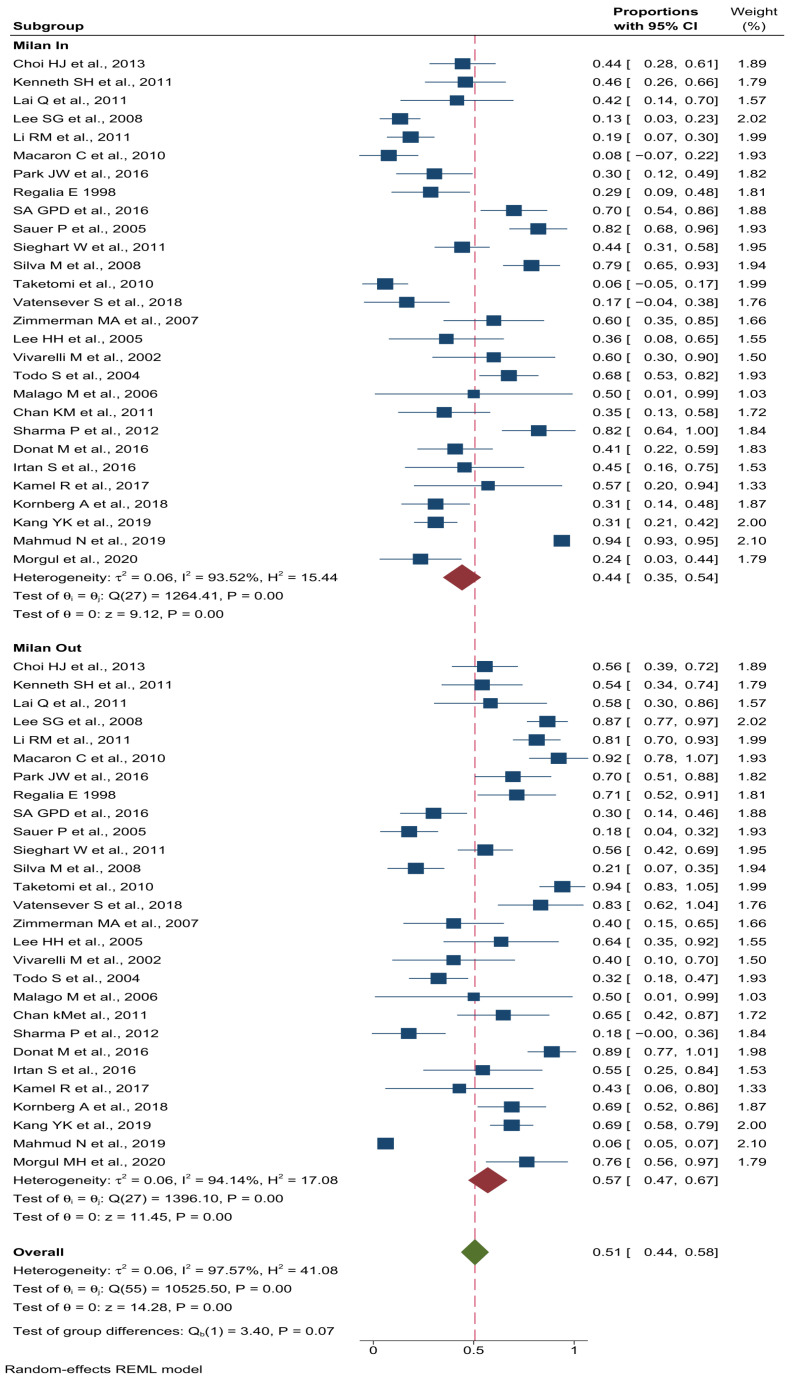
Pooled prevalence stratified by Milan criteria in patients with HCC recurrence after LT.

**Figure 4 cancers-14-05114-f004:**
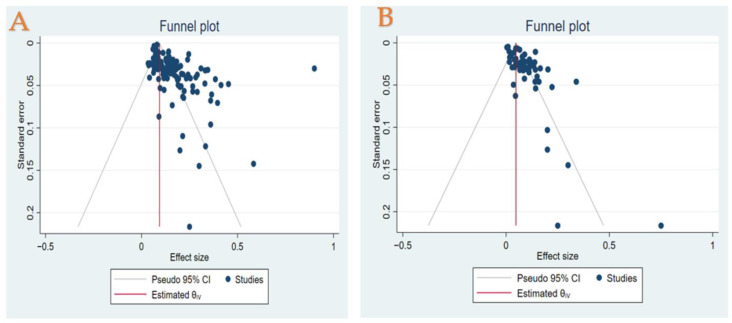
(**A**) Funnel plot of prevalence off recurrence rate in patients with HCC who underwent LT treatment. (**B**) Funnel plot of prevalence of mortality due to recurrence in patients with HCC who underwent LT treatment.

**Table 1 cancers-14-05114-t001:** Stratified met analysis of the pooled prevalence of HCC recurrence and mortality due to recurrence.

Subgroups		No of Studies	Prevalence (%) (95%CI)	Test for Heterogeneity			Between Subgroup Differences		
				Tau^2^	I^2^ (%)	*p*-Value	Q	df	*p*-Value
** *HCC Recurrence* **									
**Overall**		125	17 (15–19)	0.01	98.1	<0.001	116	1964.8	0.00
**Milan Criteria ***	Accord with	29	44 (35–54)	0.06	93.5	<0.001	1	3.40	0.00
	Do not accord with	29	57 (47–67)	0.06	97.6	<0.001			
**UCSF Criteria ***	Accord with	7	33 (18–48)	0.03	83.6	<0.001	13	35.4	0.00
	Do not accord with	7	67 (51–82)	0.03	83.6	<0.001			
**Transplant type ***	Living donor	10	19 (12–26)	0.00	0.00	0.97	9	167.6	0.00
	Deceased donor	10	81 (77–85)	0.00	0.00	0.78			
**Geographical Region**	Africa	1	12 (4–20)	0	0.00	-	4	32.00	0.00
	Asia	32	21 (18–24)	0.006	86.7	<0.001			
	Europe	55	14 (13–16)	0.004	89.2	<0.001			
	North America	28	19 (12–25)	0.029	99.6	<0.001			
	South America	9	10 (7–12)	0.001	74.4	<0.001			
**Study End Date**	1991–1999	10	23 (15–32)	0.017	91.2	<0.001	2	4.09	0.13
	2000–2009	43	18 (14–23)	0.019	95.1	<0.001			
	2010–2019	66	15 (13–17)	0.005	97.4	<0.001			
	2020–2022	6	14 (9–18)	0.002	78.3	<0.001			
**Countries Income**	Low	1	12 (4–20)	0.000	-	-	2	2.63	0.27
	Middle	25	19 (15–23)	0.011	95.1	<0.001			
	High	99	16 (14–19)	0.011	98.2	<0.001			
**Alpha-Fetoprotein** **(ng/mL)**	<50	25	14 (11–17)	0.002	90.1	<0.001	2	4.75	0.09
	≥50	13	20 (14–26)	0.001	94.3	<0.001			
**Male (%)**	<50	3	17 (13–20)	0.000	0.01	<0.001	1	0.02	0.89
	≥50	113	17 (15–19)	0.012	98.2	<0.001			
**Age (years)**	<50	11	24 (16–32)	0.017	94.4	<0.001	1	4.64	0.03
	≥50	105	16 (14–18)	0.010	97.8	<0.001			
** *Mortality in HCC Recurrence Patients* **									
**Overall**		53	9 (8–11)	0.00	88.2	<0.001	52	402.7	0.00
**Geographical Region**	Asia	12	10 (8–12)	0.001	51.8	<0.001	3	6.87	0.08
	Europe	23	10 (7–12)	0.003	88.4	<0.001			
	North America	11	11 (5–17)	0.008	97.1	<0.001			
	South America	7	7 (5–9)	0.000	38.9	<0.001			
**Countries Income**	Middle	40	9 (7–11)	0.001	71.2	<0.001	1	0.07	0.79
	High	13	9 (7–11)	0.003	90.6	<0.001			
**Study End Date**	1991–1999	5	12 (−1–24)	0.019	97.7	<0.001	3	0.96	0.81
	2000–2009	22	10 (8–12)	0.002	70.7	<0.001			
	2010–2019	21	8 (7–10)	0.001	84.7	<0.001			
	2020–2022	5	9 (6–13)	0.001	61.3	<0.001			
**Male (%)**	<50	1	7 (0–13)	0.000	-	-	1	0.66	0.42
	≥50	45	9 (8–11)	0.002	87.4	<0.001			
**Age (years)**	<50	4	18 (6–30)	0.013	96.2	<0.001	1	2.42	0.12
	≥50	42	9 (7–10)	0.002	84.3	<0.001			

*: prevalence in patients with HCC recurrence only (not all patients).

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
