# Peer review of "Hepatocellular Carcinoma Recurrence and Mortality Rate Post Liver Transplantation: Meta-Analysis and Systematic Review of Real-World Evidence"

_cancers, 2022, doi:10.3390/cancers14205114_

Round 1
Reviewer 1 Report
I read with great interest the manuscript by Khalid I. Bzeizi et al. Its an interesting metanalysis on the HCC recurrence after LT incidence across the world. The authors conclude that 17% of pts Transplanted because HCC experienced Recurrence.
I have no main concern I would only suggest to complete the literature in the discussion session making a comment on the role of the IS and IS withdrawal on the recurrence ( Manzia et al Transplant Int 2013, The Tor Vergata weaning of immunosuppression protocols in stable hepatitis C virus liver transplant patients: The 10-year follow-up; Manzia et al WJG 2019, De novo malignancies after liver transplantation: The effect of immunosuppression-personal data and review of literature). Above the well recognized prognostic factors (mRECIST, LRTs, AFP, Tumor Burden, Metrotickets 2.0 etc,).We think that the role of IS in mid and long term HCC LT recipients play an important role on the recurrence and should be much more explored.
Author Response
We thank the reviewer for the valuable comments.
The discussion has now been edited with the addition of the following (Lines 282-290):
Manzia TM et al (28), in a systematic review assessed the risk of immunosuppressive on malignancy post liver transplantation. Malignant tumor recurrence and De-Novo malignancies are the most frequent cause of mortality in adult OLT recipients with an incidence up to 26%. Calcineurin inhibitors (CNIs) seem to have a cancer-promoting influence while the mTORi might play a slight protective role reducing the incidence of post-transplant malignancy. In an earlier study, Manzia et al (29), have demonstrated the feasibility of immunosuppression withdrawal in 40% and 60% of well-selected adult and pediatric liver transplant recipients. This is likely to be a beneficial factor in minimizing HCC recurrence post-transplant

Reviewer 2 Report
I like this paper and in my opinion it should be published. I would like to congratulate the authors on their comprehensive, detailed meta-analysis and systematic review. The topic is very up-to date, the methods - appropriate and the results and conclusion - valuable. This article presents an important contribution to the worldwide discussion on the indications and role of the liver transplantation in the treatment of HCC patients. I draw my attention to very sophisticated and meticulous statistical analysis - this is an significant advantage of this paper.
I found a small, easy to correct, editorial mistake : the numbers 27. and 28. in References are identical.
Author Response
We thank to the reviewer for the kind and encouraging words.
We do apologize for the duplication of the reference and this has now been corrected. We have added two references as we have made a paragraph in the discussion section (Lines 282-290).
Kind regards
Khalid Bzeizi
